# Maestro: Uncovering Low-Rank Structures via Trainable Decomposition

## Abstract

Deep Neural Networks (DNNs) have been a large driver and enabler for AI break-throughs in recent years. These models have been getting larger in their attempt to become more accurate and tackle new upcoming use-cases, including AR/VR and intelligent assistants. However, the training process of such large models is a costly and time-consuming process, which typically yields a single model to fit all targets. To mitigate this, various techniques have been proposed in the literature, including pruning, sparsification or quantization of the model weights and updates. While able to achieve high compression rates, they often incur computational overheads or accuracy penalties. Alternatively, factorization methods have been leveraged to incorporate low-rank compression in the training process. Similarly, such tech-niques (e.g., SVD) frequently rely on the computationally expensive decomposition of layers and are potentially sub-optimal for non-linear models, such as DNNs.

In this work, we take a further step in designing efficient low-rank models and propose MAESTRO, a framework for trainable low-rank layers. Instead of regularly applying a priori decompositions such as SVD, the low-rank structure is built into the training process through a generalized variant of Ordered Dropout. This method imposes an importance ordering via sampling on the decomposed DNN structure. Our theoretical analysis demonstrates that our method recovers the SVD decomposition of linear mapping on uniformly distributed data and PCA for linear autoencoders. We further apply our technique on DNNs and empirically illustrate that MAESTRO enables the extraction of lower footprint models that preserve model performance while allowing for graceful accuracy-latency tradeoff for the deployment to devices of different capabilities.

## 1 Introduction

Deep Learning has been experiencing an unprecedented uptake, with models achieving a (super-)human level of performance in several tasks across modalities, giving birth to even more in-telligent assistants and next-gen visual perception and generation systems. However, the price of this performance is that models are getting significantly larger, with training and deployment becoming increasingly costly. Therefore, techniques from Efficient ML become evermore relevant [27], and a requirement for deployment in constrained devices, such as smartphones or IoT devices.

Typical techniques to compress the network involve *i) quantization*, i.e., reducing precision of the model [52] or communicated updates [45, 2], *ii) pruning* the model during training, e.g., through Lottery Ticket Hypothesis (LTH) [11], *iii) sparsification* of the network representation and updates, i.e., dropping the subset of coordinates [49, 3] or *iv) low-rank approximation [53, 9]*, i.e. keeping the most relevant ranks of the decomposed network. Despite the benefits during deployment, that is a lower footprint model, in many cases, the overhead during training time or the accuracy degradation

Submitted to the Workshop on Advancing Neural Network Training at 37th Conference on Neural Information Processing Systems (WANT@NeurIPS 2023). Do not distribute.

can be non-negligible. Moreover, many techniques can introduce mutliple hyperparameters or the need to fine-tune to recover the lost accuracy.

In this work, we focus on training low-rank factorized models. Specifically, we pinpoint the challenges of techniques [53, 54] when decomposing the parameters of each layer in low-rank space and the need to find the optimal ranks for each one at training time. To solve this, we adopt and non-trivially extend the Ordered Dropout technique from [17] and apply it to find progressively the optimal decomposition for each layer of a network while training (Fig. 1). Critical differences to prior work include *i)* the non-uniformity of the search space (i.e. we allow for different ranks per layer), *ii)* the trainable aspect of the decomposition to reflect the data distribution, and *iii)* the gains to training and deployment time without sacrificing accuracy. Nevertheless, we also provide a latency-accuracy trade-off mechanism to deploy the network on even more constrained devices.

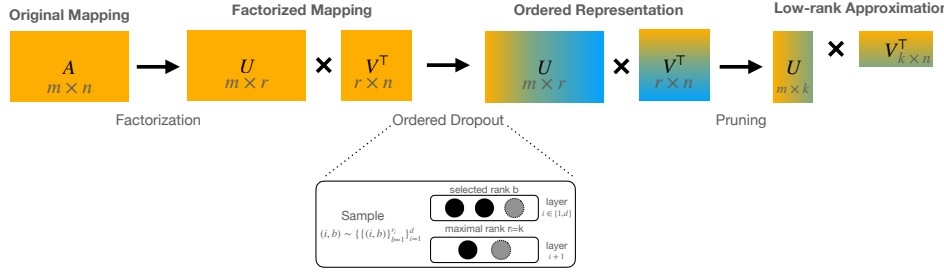

**Figure 1:** MAESTRO's construction. To obtain low-rank approximation, the given linear map is decomposed and trained with ordered dropout to obtain an ordered representation that can be efficiently pruned.

Our contributions can be summarized as follows:

- We propose MAESTRO, a novel layer decomposition technique that enables learning low-rank layers in a progressive manner while training. We novelly fuse layer factorization and an extended variant of the ordered dropout, by embedding OD directly into the factorized weights. By decomposing layers and training on stochastically sampled low-rank models, we apply ordered importance decomposed representation of each layer. We combine this with a *hierarchical group-lasso* term [64] in the loss function to zero out redundant ranks and *progressively shrink* the rank space. This way, we enable computationally efficient training achieved by the proposed decomposition without relying on inexact and potentially computationally expensive decompositions such as SVD.
- MAESTRO is a theoretically motivated approach that embeds decomposition into training. First, we show that our new objective is able to recover *i)* the SVD of the target linear mapping for the particular case of uniform data distribution and *ii)* the Principal Component Analysis (PCA) of the data in the case of identity mapping.
- As MAESTRO's decomposition is part of the training procedure, it also accounts for data distribution and the target function, contrary to SVD, which operates directly on learned weights. We show that this problem *already arises* for a simple linear model and empirically generalize our results in the case of DNNs, by applying our method to different types of layers (including fully-connected, convolutional, and attention) spanning across three datasets and modalities. We illustrate that our technique achieves better results than SVD-based baselines at a lower cost.

## 2 Related work

The topic of Efficient ML has received a lot of attention throughout the past decade as networks have been getting increasingly computationally expensive. Towards this end, we distinguish between training and deployment time, with the latter having a more significant impact and thus amortizes the potential overhead during training. Nevertheless, with the advent of Federated Learning [36], efficient training becomes increasingly relevant to remain tractable.

**Efficient inference.** For efficient deployment, there have been proposed various techniques that either optimize the architecture of the DNN in a hand-crafted [19] or automated manner (i.e. NAS) [50], they remove redundant computation by means of pruning parts of the network [12, 6, 11, 48, 30, 55, 21, 55, 65, 15, 59, 33, 62] or utilise low-precision representation [52] of the neurons and activations. Closer to our method, there have been techniques leveraging low-rank approximation (e.g. SVD) for efficient inference [58, 43, 22, 56, 9]. Last, there is a category of techniques that dynamically

resize the network at runtime for compute, memory or energy efficiency, based on early-exiting [26] or dynamic-width [63] and leverage the accuracy-latency tradeoff.

**Efficient training.** On the other hand, techniques for efficient training become very relevant nowadays when scaling DNNs sizes [20] or deploying to embedded devices [32], and oftentimes offer additional gains at deployment time. Towards this goal, there have been employed methods where part of the network is masked [46] or dropped [1, 5] during training, with the goal of minimizing the training footprint. Similarly to early-exiting, there have been proposed multi-exit variants for efficient training [24, 34], and the same applies for width-based scaling [17, 8]. Last but not least, in the era of transformers and LLMs, where networks have scaled exponentially in size, PEFT-based techniques, such as adapter-based fine-tuning [18] (such as LoRA [20]), become increasingly important and make an important differentiator for tackling downstream tasks.

**Learning ordered representation.** Originally, Ordered Dropout (OD) was proposed as a mechanism for importance-based pruning for the easy extraction of sub-networks devised to allow for heterogeneous federated training [17]. The earlier work that aims to learn ordered representation includes a similar technique to OD—Nested Dropout, which proposed a similar construction, applied to the representation layer in autoencoders [42] to enforce identifiability of the learned representation or the last layer of the feature extractor [16] to learn an ordered set of features for transfer learning. We leverage and non-trivially extend OD in our technique as a means to order ranks in terms of importance in a nested manner during training of a decomposed network that is progressively shrunk as redundant ranks converge to 0. Ranks selection is ensured through hierarchical group lasso penalty, as described in Sec. 3.3. Moreover, contrary to [17], which assumed a uniform width, our formulation allows for heterogeneous ranks per layer. Last, we leverage the ordered representation of ranks at inference time to further compress the model, allowing a graceful degradation of performance as a mechanism for the accuracy-latency trade-off.

# 3 MAESTRO

In this work, we focus on low-rank models as a technique to reduce the computational complexity and memory requirements of the neural network model. The main challenge that we face is the selection of the optimal rank or the trade-off between the efficiency and the rank for the given layer represented by linear mapping. Therefore, we devise an importance-based training technique, MAESTRO, which not only learns a mapping between features and responses, but also learns the decomposition of the trained network. This is achieved by factorizing all the layers in the network.

## 3.1 Formulation

**Low-rank approximation.** Our inspiration comes from the low-rank matrix approximation of a matrix $A \in \mathbb{R}^{m \times n}$. For simplicity, we assume that $A$ has rank $r = \min\{m, n\}$ with $k \le r$ distinct non-zero singular values $\tilde{\sigma}_1 > \tilde{\sigma}_2 > \ldots > \tilde{\sigma}_k > 0$, with corresponding left and right singular vectors $\tilde{u}_1, \tilde{u}_2, \ldots, \tilde{u}_k \in R^m$ and $\tilde{v}_1, \tilde{v}_2, \ldots, \tilde{v}_k \in R^n$, respectively. For such a matrix, we can rewrite its best $l$-rank approximation as the following minimization problem

$$\min_{U \in \mathbb{R}^{m \times l}, V \in \mathbb{R}^{n \times l}} \left\| \sum_{i=1}^{l} u_i v_i^\top - A \right\|_F^2 \tag{1}$$

where $c_i$ denotes the $i$-th row of matrix $C$ and $\|\cdot\|_F$ denotes Frobenius norm. We note that Problem (1) is non-convex and non-smooth. However, [60] showed that the randomly initialized gradient descent algorithm solves this problem in polynomial time. In this work, we consider the best rank approximation across all the ranks that leads us to the following objective

$$\min_{U \in \mathbb{R}^{m \times r}, V \in \mathbb{R}^{n \times r}} \frac{1}{r} \sum_{b=1}^{r} \left\| U_{:b} V_{:b}^\top - A \right\|_F^2, \tag{2}$$

where $C_{:b}$ denotes the first $b$ columns of matrix $C$. This objective, up to scaling, recovers SVD of $A$ exactly, and for the case of distinct non-zero singular values, the solution is, up to scaling, unique [17]. This formulation, however, does not account for the data distribution, i.e., it cannot tailor the decomposition to capture specific structures that appear in the dataset.

**Data-dependent low-rank approximation.** Therefore, the next step of our construction is to extend this problem formulation with data that can further improve compression, reconstruction, and generalization, and incorporate domain knowledge. We assume that data comes from the distribution

$x \sim \mathcal{X}$ centered around zero, i.e., $\mathbf{E}_{x \sim \mathcal{X}}[x] = 0$.[1], and the response is given by $y = Ax$. In this particular case, we can write the training loss as

$$\min_{U \in \mathbb{R}^{m \times r}, V \in \mathbb{R}^{n \times r}} \mathbf{E}_{x,y \sim \mathcal{X}} \left[ \sum_{b=1}^{r} \frac{1}{r} \left\| U_{:b} V_{:b}^{\top} x - y \right\|^2 \right]. \tag{3}$$

It is important to note that the introduced problem formulation (3) is the same as the Ordered Dropout formulation of [17] for the neural network with a single hidden layer and no activations, and it can be solved using stochastic algorithms by sampling from the data distribution $\mathcal{X}$ (subsampling) and rank distribution $\mathcal{D}$. However, there is an important distinction when we apply MAESTRO for deep neural networks. While FjORD applies uniform dropout across the width of the network for each layer, we propose to decompose each layer independently to uncover its – potentially different – optimal rank for deployment. We discuss details in the next paragraph.

**DNN low-rank approximation.** For Deep Neural Networks (DNNs), we seek to uncover the optimal ranks for a set of $d$ linear mappings $W^1 \in \mathbb{R}^{m_1 \times n_1}, \ldots, W^d \in \mathbb{R}^{m_d \times n_d}$, where $W^i$'s are model parameters and $d$ is model depth, e.g., weights corresponding to linear layers[2], by decomposing them as $W^i = U^i \left(V^i\right)^{\top}$. We discuss how these are selected in the next section. To decompose the network, we aim to minimize the following objective:

$$\mathbf{E}_{x,y \sim \mathcal{X}} \left[ \frac{1}{\sum_{i=1}^{d} r_i} \sum_{i=1}^{d} \sum_{b=1}^{r_i} l(h(U^1\left(V^1\right)^{\top}, \ldots, U_{:b}^i\left(V_{:b}^i\right)^{\top}, \ldots, U^d\left(V^d\right)^{\top}, W^o, x), y) \right], \tag{4}$$

where $r_i = \min\{m_i, n_i\}$, $l$ is a loss function, $h$ is a DNN, and $W^o$ are the other weights that we do not decompose. We note that our formulation aims to decompose each layer, while decompositions across layers do not directly interact. The motivation for this approach is to uncover low-rank structures within each layer that are not affected by inaccuracies from other layers due to multiple low-rank approximations.

## 3.2 Layer factorization

The following subsections discuss how model factorization is implemented for different model architectures.

**FC layers.** A 2-layer fully connected (FC) neural network can be expressed as $f(x) = \sigma(\sigma(xW_1)W_2)$, where $W$s are weight matrices of each FC layer, and $\sigma(\cdot)$ is any arbitrary activation function, e.g., ReLU. The weight matrix $W$ can be factorized as $UV^{\top}$.

**CNN layers.** For a convolution layer with dimension, $W \in \mathbb{R}^{m \times n \times k \times k}$ where $m$ and $n$ are the number of input and output channels, and $k$ is the size of the convolution filters. Instead of directly factorizing the 4D weight of a convolution layer, we factorize the unrolled 2D matrix. Unrolling the 4D tensor $W$ leads to a 2D matrix with shape $W_{\text{unrolled}} \in \mathbb{R}^{mk^2 \times n}$, where each column represents the weight of a vectorized convolution filter. Factorization can then be conducted on the unrolled 2D matrix; see [53] for details.

**Transformers.** A Transformer layer consists of a stack of encoders and decoders [51]. The encoder and decoder contain three main building blocks: the multi-head attention layer, position-wise feed-forward networks (FFN), and positional encoding. We factorize all trainable weight matrices in the multi-head attention (HMA) and the FFN layers. The FFN layer factorization can directly adopt the strategy from the FC factorization. A $p$-head attention layer learns $p$ attention mechanisms on the key, value, and query $(K, V, Q)$ of each input token:

$$\text{MHA}(Q, K, V) = \text{Concat}(\text{head}_1, \ldots, \text{head}_p)W^O.$$

Each head performs the computation of:

$$\text{head}_i = \text{Attention}(QW_Q^{(i)}, KW_K^{(i)}, VW_V^{(i)}) = \text{softmax}\left( \frac{QW_Q^{(i)}W_K^{(i)\top}K^{\top}}{\sqrt{d/p}} \right) VW_V^{(i)}.$$

where $d$ is the hidden dimension. The trainable weights $W_Q^{(i)}, W_K^{(i)}, W_V^{(i)}, i \in \{1, 2, \ldots, p\}$ can be factorized by simply decomposing all learnable weights $W_\cdot$ in an attention layer and obtaining $U \cdot V^{\top}$ [51].

---

[1] We make this assumption for simplicity. It can be simply overcome by adding a bias term into the model.

[2] We can apply our decomposition on different types of layers, such as Linear, Convolutional and Transformers as shown in Sec. 3.2.

### 3.3 Training techniques

Having defined the decomposition of typical layers found in DNNs, we move to formulate the training procedure of our method, formally described in Algorithm 1. Training the model comprises an iterative process of propagating forward on the model by *sampling a rank* $b_i$ per decomposed layer $i$ up to maximal rank $r_i$ (line 3). We calculate the loss, which integrates an additional *hierarchical group lasso* component (lines 4) and *backpropagate* on the sampled decomposed model (line 5). At the end of each epoch, we *progressively shrink* the network by updating the maximal rank $r_i$, based on an importance threshold $\varepsilon_{ps}$ (line 11). We provide more details about each component below.

---

**Algorithm 1:** MAESTRO (Training Process)

**Input:** epochs $E$, dataset $\mathcal{D}$, model $h$ parametrized by $U^1 \in \mathbb{R}^{m_1 \times r_1}$,
$\quad\quad V^1 \in \mathbb{R}^{n_1 \times r_1}, \ldots, U^d \in \mathbb{R}^{m_d \times r_d}, V^d \in \mathbb{R}^{n_d \times r_d}, W^o$, and hyperparameters $\lambda_{gl}, \varepsilon_{ps}$

1   **for** $t \leftarrow 0$ **to** $E - 1$ **do** // *Epochs*
2     **for** $(x, y) \in \mathcal{D}$ **do** // *Iterate over dataset*
3       Sample $(i, b) \sim \left\{ \{(i, b)\}_{b=1}^{r_i} \right\}_{i=1}^{d}$;
4       $L = l(h(U^1{\left(V^1\right)}^\top, \ldots, U^i_{:b}{\left(V^i_{:b}\right)}^\top, \ldots, U^d{\left(V^d\right)}^\top, W^o, x), y)+$
        $+\lambda_{gl} \sum_{i=1}^{d} \sum_{b=1}^{r_i} (\|U^i_{b:}\| + \|V^i_{b:}\|)$ // *compute loss*
5       L.backward() // *Update weights*
6     **end**
7     **for** $i \leftarrow 1$ **to** $d$ **do**
8       **for** $b \leftarrow 1$ **to** $r_i$ **do**
9         // *rank importance thresholding*
10        **if** $\|V^i_{b:}\|\|U^i_{b:}\| \leq \varepsilon_{ps}$ **then**
11          $r_i = b - 1$ // *progressive shrinking*
12          **break**
13        **end**
14       **end**
15     **end**
16 **end**

---

**Efficient training via sampling.** In Sec. 4, we show that for the linear case (3), the optimal solution corresponds to PCA over the linearly transformed dataset. This means that the obtained solution contains orthoghonal directions. This property is beneficial because it directly implies that when we employ gradient-based optimization, not only is the gradient zero at the optimum, but the gradient with respect to each summand in Equation (3) is also zero. The same property is directly implied by overparametrization [35] or strong growth condition [44]. As a consequence, this enables us to sample only one summand at a time and obtain the same quality solution. When considering (4) as an extension to (3), it is unclear whether this property still holds, which would also imply that the set of stationary points of (3) is a subset of stationary points of the original objective without decomposition. However, in the experiments, we observed that sampling is sufficient to converge to a good-quality solution. If this only holds approximately, we one could leverage fine-tuning to recover the loss in performance.

**Efficient rank extraction via hierarchical group-lasso.** By definition, (3) leads to an ordered set of ranks for each layer. This ordered structure enables efficient rank extraction and selection. To effectively eliminate unimportant ranks while retaining the important ones, thus leading to a more efficient model, we consider Hierarchical Group Lasso (HGL) [31] in the form

$$\lambda_{gl} \sum_{i=1}^{d} \sum_{b=1}^{r_i} (\|U^i_{b:}\| + \|V^i_{b:}\|), \tag{5}$$

where $C_{b:}$ denotes the matrix that contains all the columns of $C$ except for the first $b - 1$ columns.

**Progressive shrinking.** HGL encourages that unimportant ranks become zero and can be effectively removed from the model. To account for this, for each layer we remove $V^i_{b:}$ and $U^i_{b:}$ (i.e., set $r_i = b-1$) if $\|V^i_{b:}\|\|U^i_{b:}\| \leq \varepsilon_{ps}$, where $\varepsilon_{ps}$ is a pre-selected threshold – and a hyperparameter of our method.

**Initialization.** Initialization is a key component of the training procedure [13, 37]. To adopt the best practices from standard non-factorized training, we follow a similar approach to [23, 53], where we first initialize the non-factorized model using standard initialization. For initializing factorized layers, we use the Singular Value Decomposition (SVD) of the non-factorized initialization – in a

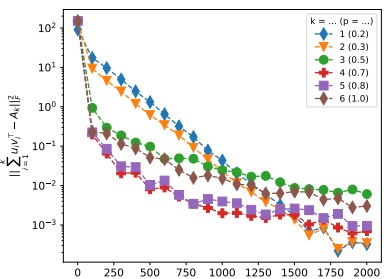
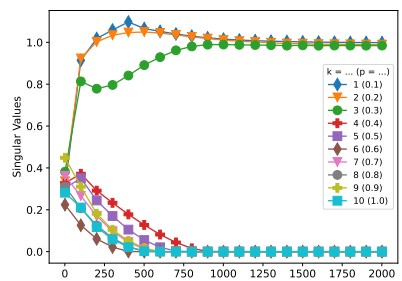

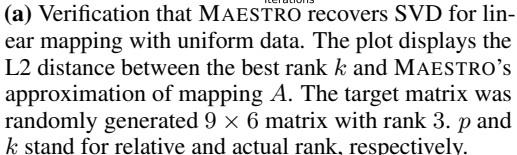

**(a)** Verification that MAESTRO recovers SVD for linear mapping with uniform data. The plot displays the L2 distance between the best rank $k$ and MAESTRO's approximation of mapping $A$. The target matrix was randomly generated $9 \times 6$ matrix with rank 3. $p$ and $k$ stand for relative and actual rank, respectively.

**(b)** Verification that MAESTRO recovers PCA for identity mapping. The plot displays the estimates of singular values. The data distribution has only 3 directions. It is expected that the top 3 ranks will converge to value one and the rest to zero. $p$ and $k$ stand for relative and actual rank, respectively.

**Figure 2:** Empirical showcase of theoretical properties of the MAESTRO's formulation.

full-rank form – to ensure that the resulting product matrix is the same as the original parameter decomposition. In addition, SVD is an optimal decomposition for the linear case with uniform data. However, in contrast with the adaptive baseline method [54] we only decompose once, rather than on every training iteration.

### 3.4 Train-once, deploy-everywhere

Up until now, we have described how our method works for training low-rank models, which yield computational, memory, network, and energy [57] bandwidth benefits during training. At deployment time, one can directly deploy the final model (rank $r_i$ for each layer) on the device, which we acquire from performing a threshold sweep of $\varepsilon_{ps}$ over the effective range of rank importance across layers. However, in case we want to run on even more constrained devices, such as mobile [4] or embedded [4] systems, the learned decomposition also gives us the flexibility to further compress the model in a straightforward manner, effectively trading off accuracy for a smaller model footprint. Inspired by [61], we propose to use greedy search. We begin with the current model and compare model performance across various low-rank models, each created by removing a certain percentage of ranks from each layer. We then eliminate the ranks that cause the least decrease in performance. This process is iterated until we reach the desired size or accuracy constraint. To make this approach efficient, we estimate the loss using a single mini-batch with a large batch size, for example, 2048. This also avoids issues with BatchNorm layers; see [61] for details.

In summary, MAESTRO comprises a technique for trainable low-rank approximation during training time that progressively compresses the model, reflecting the data distribution, and a method that enables a graceful trade-off between accuracy and latency for embedded deployment, by selecting the most important parts of the network. We validate these claims in Sec. 5.2 and 5.5, respectively.

## 4 Theoretical guarantees

In this section, we further investigate the theoretical properties of MAESTRO for the linear mappings, i.e., the setup of the problem formulation (3).

**Theorem 4.1** (Informal). *Let $A = \tilde{U}\tilde{\Sigma}\tilde{V}^\top$ be a SVD decomposition of A. Then, the minimization problem (3) is equivalent to PCA applied to the transformed dataset $x \to \tilde{\Sigma}\tilde{V}^\top x$, $x \sim \mathcal{X}$ projected on the column space of $\tilde{U}$.*

The formal statement can be found in Appendix D. Theorem 4.1 shows that MAESTRO can adapt to data distribution by directly operating on data $x \sim \mathcal{X}$ and also to the target mapping by projecting data to its right singular vectors scaled by singular values. In particular, we show that in the special case, when $\mathcal{X}$ is the uniform distribution on the unit ball, (3), i.e., MAESTRO, exactly recovers truncated SVD of $A$, which is consistent with the prior results [17]. In the case $A$ is the identity, it is straightforward to see that MAESTRO is equivalent to PCA. We can see that MAESTRO can efficiently extract low-rank solutions by filtering out directions corresponding to the null space of the target mapping $A$ and directions with no data. We also numerically verify both of the special cases–PCA and SVD, by minimizing (3) using stochastic gradient descent (SGD) with $\mathcal{D}$ being the uniform distribution. These preliminary experiments are provided in Fig. 2a and 2b.

We showed that MAESTRO could recover SVD in a particular case of the linear model and the uniform data distribution on the unit ball. We note that in this case, SVD is optimal, and we cannot acquire better decomposition. Therefore, it is desired that MAESTRO is equivalent to SVD in this scenario. In the more general setting, we argue that MAESTRO decomposition should be preferable to SVD due to the following reasons:

- MAESTRO formulation is directly built into the training and tailored to obtain the best low-rank decomposition, while SVD relies on linearity assumption.
- SVD does not account for data, and even in the linear NN case, the learned singular vectors might exhibit wrong ordering. We demonstrate this issue using a simple example where we take matrix $A$ with rank 3. We construct the dataset $\mathcal{X}$ in such a way that the third singular vector is the most important, the second one is the second, and the first is the third most important direction. Clearly, SVD does not look at data. Therefore, it cannot capture this phenomenon. We showcase that MAESTRO learns the correct order; see Fig. 5 of the Appendix.
- Pre-factorizing models allow us to apply hierarchical group-lasso penalty [64] for decomposed weights to directly regularize the rank of different layers.
- SVD is computationally expensive and can only run rarely, while MAESTRO is directly built into the training and, therefore, does not require extra computations. In addition, MAESTRO supports rank sampling so training can be made computationally efficient.

## 5   Experiments

We start this section by describing the setup of our experiments, including the models, datasets and baselines with which we compare MAESTRO. We then compare MAESTRO against the baselines on accuracy and MAC and discuss the results. Subsequently, we analyze the behaviour of our system in-depth and provide additional insights on the performance of our technique, along with an ablation study and sensitivity analysis to specific hyperparameters. Finally, we showcase the performance of models upon deployment and how we can derive a smaller footprint model with some accuracy trade-off, without the need to fine-tune.

### 5.1   Experimental setup

**Models & datasets.** The datasets and models considered in our experiments span across four datasets, concisely presented along with the associated models on Tab. 1. We have implemented our solution with PyTorch [38](v1.13.0) trained our models on NVidia A100 (40G) GPUs. Details for the learning tasks and hyperparameters used are presented in the Appendix.

**Baselines.** We have selected various baselines from the literature that we believe are closest to aspects of our system. On the *pruning* front, we compare with the IMP [40] and RareGems [48] techniques, themselves based on the LTH [11]. On the *quantization* front, we compare with XNOR-Net [41]. With respect to *low-rank* methods, we compare with Spectral Initialisation [23], Pufferfish [53] and Cuttlefish [54].

**Table 1:** Datasets and models for evaluation. The network footprints depict the vanilla variants of the models.

| Dataset | Model | # GMACs | # Params (M) | Task |
|---|---|---|---|---|
| **MNIST** | LeNet | $2e^{-4}$ | 0.04 | Image classification |
| **CIFAR10** | ResNet-18 | 0.56 | 11.18 | Image classification |
| **CIFAR10** | VGG-19 | 0.40 | 20.00 | Image classification |
| **TinyImageNet** | ResNet-50 | 5.19 | 53.9 | Image classification |
| **Multi30k** | 6-layer Transformer | 1.37 | 48.98 | Translation (en-ge) |

### 5.2   Performance comparison

We start off by comparing MAESTRO with various baselines from the literature across different datasets and types of models[3]. Results are depicted in Tab. 2 and 3, while additional performance points of MAESTRO for different model footprints are presented in the Appendix F.2 and F.3.

**Comparisons with low-rank methods.** The low-rank methods we are comparing against are Pufferfish [53] and Cuttlefish [54]. These methods try to reduce training and inference runtime while preserving model accuracy by leveraging low-rank approximations. For ResNet-18, we achieve $94.19\pm0.07\%$ for 4.08M parameters and $93.97\pm0.25\%$ for 2.19M parameters compared to the $94.17\%$ of Pufferfish at 3.3M parameters. For VGG-19, we achieve +0.41pp (percentage points) higher accuracy compared to Pufferfish and -0.29pp to Cuttlefish at 44.8% and 93.2% of the sizes,

---

[3]The operating points we select for MAESTRO are the closest lower to the respective baseline in terms of footprint. Where the result is not present in the Tab. 2, we provide the $\lambda_{gp}$ value so that it can be referenced from the Appendix, Tab. 11, 12.

**Table 2:** Maestro vs. baselines on CIFAR10.

| Variant | Model | Acc. (%) | GMACs | Params. ($M$) |
|---|---|---|---|---|
| Non-factorized | ResNet-18 | $93.86_{\pm0.20}$ | 0.56 | 11.17 |
| Pufferfish | ResNet-18 | 94.17 | 0.22 | 3.336 |
| Cuttlefish | ResNet-18 | 93.47 | 0.3 | 3.108 |
| IMP | ResNet-18 | 92.12 | - | 0.154 |
| RareGems | ResNet-18 | 92.83 | - | **0.076** |
| XNOR-Net | ResNet-18 | 90.06 | - | $0.349^{\dagger}$ |
| MAESTRO$^{\dagger}$ ($\lambda_{gp}=16e^{-6}$) | ResNet-18 | $94.19_{\pm0.07}$ | $0.39_{\pm0.00}$ | $4.08_{\pm0.02}$ |
| MAESTRO$^{\dagger}$ ($\lambda_{gp}=64e^{-6}$) | ResNet-18 | $93.86_{\pm0.11}$ | $0.15_{\pm0.00}$ | $1.23_{\pm0.00}$ |
| Non-factorized | VGG-19 | $92.94_{\pm0.17}$ | 0.40 | 20.56 |
| Pufferfish | VGG-19 | 92.69 | 0.29 | 8.37 |
| Cuttlefish | VGG-19 | **93.39** | 0.15 | 2.36 |
| RareGems | VGG-19 | 86.28 | - | 5.04 |
| IMP | VGG-19 | 92.86 | - | 5.04 |
| XNOR-Net | VGG-19 | 88.94 | - | $0.64^{\dagger}$ |
| Spectral Init.$^{*}$ | VGG-19 | 83.27 | - | $\approx 0.4$ |
| MAESTRO$^{\dagger}$ ($\lambda_{gp}=32e^{-6}$) | VGG-19 | $93.10_{\pm0.10}$ | $0.13_{\pm0.00}$ | $2.20_{\pm0.03}$ |
| MAESTRO$^{\dagger}$ ($\lambda_{gp}=512e^{-6}$) | VGG-19 | $88.53_{\pm0.13}$ | $\mathbf{0.03_{\pm0.00}}$ | $\mathbf{0.35_{\pm0.00}}$ |

$^{*}$Results from original work; $^{\dagger}$ XNOR-Net employs binary weights and activations; although the overall #trainable parameters remain the same as the vanilla network, each model weight is quantized from 32-bit to 1-bit. Therefore, we report a compression rate of $3.125\%(^{1}/_{32})$.

**Table 3:** Maestro vs. baselines on Multi30k.

| Variant | Model | Perplexity | GMACs | Params. ($M$) |
|---|---|---|---|---|
| Non-factorized | Transformer | $9.85_{\pm0.10}$ | 1.370 | 53.90 |
| Pufferfish$^{*}$ | Transformer | $7.34_{\pm0.12}$ | 0.996 | 26.70 |
| MAESTRO$^{\dagger}$ | Transformer | $\mathbf{6.90_{\pm0.07}}$ | $\mathbf{0.248_{\pm0.0032}}$ | $\mathbf{13.80_{\pm0.113}}$ |

$^{*}$Results from original work; $^{\dagger}$ tuned $\lambda_{gp}$ from $\{2^{i}/100; i \in 0, \ldots, 9\}$

**Table 4:** Ablation study for ResNet18 on CIFAR10

| Variant | Acc. (%) | GMACs | Params. ($M$) |
|---|---|---|---|
| MAESTRO | $94.19_{\pm0.39}$ | $0.39_{\pm0.0008}$ | $4.08_{\pm0.020}$ |
| **w/out GL** | $94.04_{\pm0.10}$ | $0.56_{\pm0.0000}$ | $11.2_{\pm0.000}$ |
| **w/out PS** | $94.12_{\pm0.36}$ | $0.39_{\pm0.0010}$ | $4.09_{\pm0.027}$ |
| **w/ full-training** | $94.05_{\pm0.32}$ | $0.39_{\pm0.0004}$ | $4.09_{\pm0.032}$ |

respectively. Finally, comparing with the spectral initialization [23] for VGG-19, we achieve +5.26pp higher accuracy for 87.5% of parameter size. Detailed results are shown in Tab. 2. This performance benefits also apply in the case of Transformers (Tab. 3), where MAESTRO performs 6% better in terms of perplexity at 25% of the cost (MACs) and 51.7% of the size (parameters) compared to Pufferfish.

**Comparisons with pruning methods.** The next family of baselines is related to the LTH [11]. Specifically, we compare against IMP [40] and witness from Tab. 2 that MAESTRO can achieve +1.25pp ($\lambda_{gp}=128e^{-6}$) and +0.24pp ($\lambda_{gp}=32e^{-6}$) higher accuracy for ResNet-18 and VGG-19 respectively. Although we cannot scale to the size that RareGems [48] for ResNet-18, the sparsity that they achieve is unstructured, which most modern hardware cannot take advantage of. In contrast, our technique performs ordered structured sparsity, compatibly with most computation targets. On the other hand, for VGG-19, we achieve +6.82pp higher accuracy at 43.6% of the footprint.

**Comparisons with quantized models.** We also compare against XNOR-Net [41], which binarizes the network to achieve efficient inference. Training continues to happen in full precision, and inference performance is really dependent on the operation implementation of the target hardware. Nonetheless, assuming a compression rate of 3.125%, for the same size of network, we achieve +1.08pp ($\lambda_{gp}=512e^{-6}$) and +2.18pp ($\lambda_{gp}=256e^{-6}$) higher accuracy on ResNet-18 and VGG-19.

### 5.3 Training behaviour of MAESTRO

Having shown the relative performance of our framework to selected baselines, we now move to investigate how our method behaves, with respect to its convergence and low-rank approximations.

**Model and rank convergence.** In Fig. 3, we present the training dynamics for MAESTRO. Fig. 3a illustrates the evolution of total rank throughout the training steps. We observe that the ranks are pruned incrementally. This aligns with the observations made during Pufferfish [53] training, where the authors suggest warm-start training with full precision to enhance the final model performance. In our situation, we do not need to integrate this heuristic because MAESTRO automatically prunes rank. Fig. 3b reveals the ranks across layers after training. We notice an intriguing phenomenon: the ranks are nested for increasing $\lambda_{gl}$. This could imply apart from a natural order of ranks within each layer, a global order. We briefly examine this captivating occurrence in the following section, and we plan to investigate it more thoroughly in future work, as we believe this might contribute to a superior rank selection and sampling process. Lastly, Fig. 3c depicts the progression of training loss. We find that our hypothesis, that sampling does not adversely impact training, is also supported empirically.

### 5.4 Ablation study

In this section, we examine the impact of each component on the performance of MAESTRO. Specifically, we run variants of our method *i)* without the *hierarchical group lasso regularization (HGL)*, *ii)* without progressive *shrinking (PS)*. Additionally, we integrate *iii)* an *extra full low-rank pass* ($b = r_i$) into the training at each step to assess whether extra sampling would be beneficial. The results are displayed in Tab. 4. As anticipated, our findings confirm that neither inclusion of hierarchical group lasso with a tuned $\lambda_{gl}$ nor progressive shrinking impair the final performance,

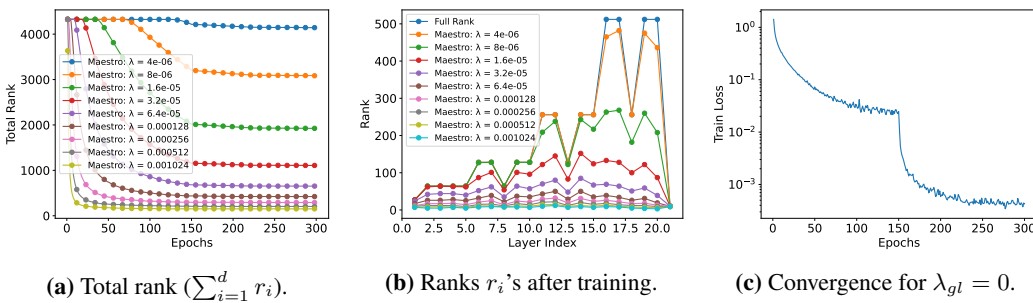

**(a)** Total rank ($\sum_{i=1}^{d} r_i$).  **(b)** Ranks $r_i$'s after training.  **(c)** Convergence for $\lambda_{gl} = 0$.

**Figure 3:** Training dynamics of MAESTRO for ResNet18 on CIFAR10.

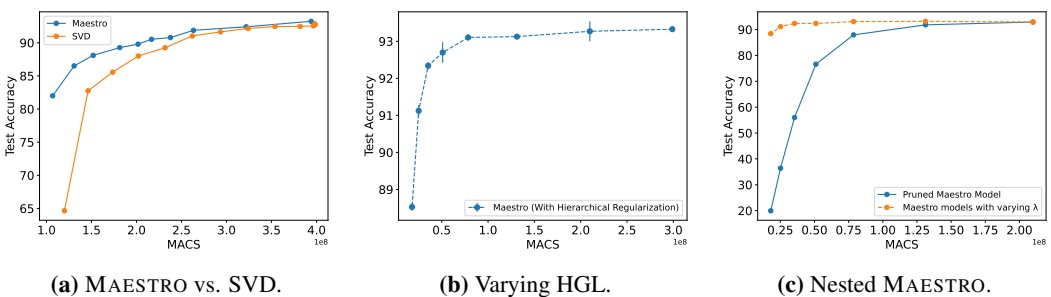

**(a)** MAESTRO vs. SVD.  **(b)** Varying HGL.  **(c)** Nested MAESTRO.

**Figure 4:** Accuracy-latency trade-off of MAESTRO under different settings for VGG19 on CIFAR10.

but they do significantly enhance the efficiency of MAESTRO. Moreover, sampling more ranks at each training step does not improve the final performance, and, in fact, it hampers training efficiency, making it approximately twice as computationally demanding.

## 5.5 Accuracy-latency trade-off at training and deployment time

In Fig. 4, we illustrate various approaches to balance latency (proxied through MACs operations) and accuracy in model training and deployment. Fig. 4a demonstrates how MAESTRO ($\lambda_{gl} = 0$) can be pruned effectively for deployment using the greedy search method discussed in Section 3.4. We contrast this with the greedy pruning of a non-factorized model that has been factorized using SVD. We reveal that this straightforward baseline does not measure up to the learned decomposition of MAESTRO and results in a significant performance decrease. Next, Fig. 4b portrays the final accuracy and the number of model parameters for varying hierarchical group lasso penalties. This leads to the optimal latency-accuracy balance for both training and inference. However, it's crucial to point out that each model was trained individually, while greedy pruning only necessitates a single training cycle. Lastly, we delve into the observation of nested ranks across increasing $\lambda_{gl}$. Fig. 4c displays the performance of MAESTRO ($\lambda_{gl} = 0$) across different ranks selected by smaller models MAESTRO ($\lambda_{gl} > 0$). Intriguingly, we observe that MAESTRO ($\lambda_{gl} = 0$) performs very well—for instance, we can decrease its operations in half (and parameters by $10\times$) and still maintain an accuracy of $87.7\%$ without fine-tuning, just by reusing rank structure from independent runs. As aforementioned, we intend to further explore this in the future.

## 6 Conclusion and future work

In this work, we have presented MAESTRO, a method for trainable low-rank approximation of DNNs that leverages progressive shrinking by applying a generalized variant of Ordered Dropout to the factorized weights. We have shown the theoretical guarantees of our work in the case of linear models and empirically demonstrated its performance across different types of models, datasets, and modalities. Our evaluation has demonstrated that MAESTRO outperforms competitive compression methods at a lower cost. In the future, we plan to expand our technique to encompass more advanced sampling techniques and apply it to different distributed learning scenarios, such as Federated Learning, where data are natively non-independent or identically distributed (non-IID).

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

# Appendix

## Contents of the Appendix

## A   Broader impact

The goal of our work is to make the training and deployment of DNNs more efficient, affecting the total computation, memory and bandwidth of systems, as well as the energy they require to run the respective tasks. DNN model training requires significant amounts of energy, whether in a data center or at the edge [57, 39]. However, such techniques should not be used as an excuse to make data centers less green, but rather as a complementary measure to further reduce the carbon footprint of Deep Learning.

Additionally, as our technique involves a training-aware methodology for progressively selecting ranks, it depends on the quality of data used in training. Deploying the model in the wild for various downstream tasks may result in behavior different from the intended outcomes. Therefore, it should be thoroughly tested before deployment to ensure it adheres to the required Service Level Objectives (SLOs), especially in performance-critical use cases, such as self-driving vehicles or UAV navigation.

## B   Limitations

In this work, we have proposed a method for trainable low-rank approximation of DNNs that provides performance benefits for both training and inference times. While we suggest that this could have repercussions on the energy consumption of these tasks, we have not yet evaluated this hypothesis experimentally across different devices, be they data center-grade or at the edge.

Additionally, we have applied our technique to CNN and Transformer models spanning across vision and NLP tasks. While we anticipate generalization to any type of network, it remains to be seen whether our techniques can also be applied to alternative types of layers, such as recurrent ones, and the benefits they may bring.

Although we have provided a thorough investigation of the behaviour of our proposed system, the only way we can control the end footprint of the model during training is via the $\lambda_{gl}$ and $\varepsilon_{ps}$ hyperparameters. However, there is no guarantee about the final footprint of the model. If we are willing to sacrifise accuracy, then the technique illustrated in Sec. 3.4 and evaluated in Sec. 5.5 is a start. More robust ways of globally ranking per-layer importances are left as future work.

Lastly, our sampling method during training is uniform up to the maximum rank during progressive shrinking. Although this method has proven effective, alternative sampling methods could potentially

accelerate rank exploration, thereby hastening the shrinking and convergence of the network during training.

## C  Extended Background

**Ordered Dropout.** Ordered Dropout is a technique of importance-based, nested and ordered pruning that works along the indices of a layer's parameters (neurons, filters, etc.) Introduced by [17], the authors describe a training technique where a layer's width is discretised in $|P|$ values, where $P = \{s_1, s_2, \ldots, s_{|P|}\}$, and at each training step, they sample $p \sim U_P$ to get a specific subnetwork, extracted by selecting the first $\lceil p * K_l - 1 \rceil$ neurons per layer and dropping the rest. In contrast to our work, sampling is happening directly on model parameters (rather than ranks) and is uniform across layers (i.e. a single p-value is set). Nested-ness refers to the fact that larger p-value models include the parameters of lower p-values and importance-based pruning means that via stochastic sampling, the right-most (in terms of index) parameters train on progressively less data due to the probability of sampling and nestedness (i.e. all data pass from the parameters of minimal subnetwork, less pass the higher the p-value).

## D  Theoretical Properties of Low-Rank Layers

In this section, we show that for the case of linear mappings, i.e., the problem formulation discussed in (3), MAESTRO acts as PCA applied to the original dataset $\mathcal{X}$ projected onto the space weighted by the corresponding singular values. Before proceeding with the theorem, we first recall the assumptions and notations introduced in the main paper.

We denote $C_{:b}$ as the first $b$ columns of matrix $C$, $C_{:a,:b}$ denotes the first $a$ rows, and $b$ columns of a matrix $C$, $a+1 :$ denotes the all the columns/rows from index $a+1$, : denotes the all the columns/rows, and for vectors, we use a single subscript. As discussed in the main paper, we reformulate the original least squares problems to the following decomposition problem

$$\min_{U \in \mathbb{R}^{m \times r}, V \in \mathbb{R}^{n \times r}} \mathbf{E}_{x,y \sim \mathcal{X}} \left[ \mathbf{E}_{b \sim \mathcal{D}} \left[ \left\| U_{:b} V_{:b}^\top x - y \right\|^2 \right] \right], \tag{6}$$

where $\mathcal{D}$ is a distribution that samples $b \in \{1, 2, \ldots, r\}$ with probability $p_b > 0$ and we assume that $y$ is linked with $x$ through linear map $A$, i.e., $y = Ax$.

**Theorem D.1.** *Let $A = \tilde{U}\tilde{\Sigma}\tilde{V}^\top$ be a SVD decomposition of $A$. Then, the minimization problem* (6) *is equivalent to PCA applied to the transformed dataset $x \to \tilde{\Sigma}\tilde{V}^\top x$, $x \sim \mathcal{X}$ projected on the column space of $\tilde{U}$. Concretely, we can first solve*

$$\min_{U \in \mathbb{R}^{m \times r}, V \in \mathbb{R}^{n \times r}} \mathbf{E}_{z \sim \mathcal{X}} \left[ \mathbf{E}_{b \sim \mathcal{D}} \left[ \left\| \left( U_{:b} V_{:b}^\top - I \right) \tilde{\Sigma}\tilde{V}^\top x \right\|^2 \right] \right], \tag{7}$$

*and then we can obtain the solutions of* (6) *using $U^\star = \tilde{U}^\top \bar{U}$, $V^\star = \tilde{V}^\top \bar{V}$, where $\bar{U}, \bar{V}$ belong to the set of optimal solutions of problem* (7).*zx*

*In the particular case, where $\mathcal{X}$ is a uniform distribution on the unit ball,* (6) *recovers the best rank approximation of $A$ across all ranks, i.e., up to the scale of $U$ and $V$ recovers truncated SVD. In the case, $A$ is identity,* (6) *leads to standard PCA decomposition.*

*Proof.* From the assumptions that $y = Ax$ and $A = \tilde{U}\tilde{\Sigma}\tilde{V}^\top$, we can rewrite (6) as

$$\min_{U \in \mathbb{R}^{m \times r}, V \in \mathbb{R}^{n \times r}} \mathbf{E}_{x \sim \mathcal{X}} \left[ \mathbf{E}_{b \sim \mathcal{D}} \left[ \left\| \left( U_{:b} V_{:b}^\top - \tilde{U}\tilde{\Sigma}\tilde{V}^\top \right) x \right\|^2 \right] \right].$$

Since $\tilde{U}$ is orthogonal, we have $\|z\| = \|\tilde{U}^\top z\|$. Therefore, the above problem is equivalent to

$$\min_{U \in \mathbb{R}^{m \times r}, V \in \mathbb{R}^{n \times r}} \mathbf{E}_{x \sim \mathcal{X}} \left[ \mathbf{E}_{b \sim \mathcal{D}} \left[ \left\| \left( \tilde{U}^\top U_{:b} V_{:b}^\top - \tilde{\Sigma}\tilde{V}^\top \right) x \right\|^2 \right] \right],$$

which is also equivalent to

$$\min_{U\in\mathbb{R}^{m\times r},V\in\mathbb{R}^{n\times r}} \mathbf{E}_{x\sim\mathcal{X}}\left[\mathbf{E}_{b\sim\mathcal{D}}\left[\left\|\left(U_{:b}V_{:b}^{\top}-\tilde{\Sigma}\tilde{V}^{\top}\right)x\right\|^{2}\right]\right]$$

after reparametrization. The next step involves injecting identity in the form $\tilde{V}\tilde{V}^{\top}$ as that leads to the equivalent reformulation

$$\min_{U\in\mathbb{R}^{m\times r},V\in\mathbb{R}^{n\times r}} \mathbf{E}_{x\sim\mathcal{X}}\left[\mathbf{E}_{b\sim\mathcal{D}}\left[\left\|\left(U_{:b}V_{:b}^{\top}\tilde{V}-\tilde{\Sigma}\right)\tilde{V}^{\top}x\right\|^{2}\right]\right].$$

As for the previous case, we can reparametrise the problem to obtain

$$\min_{U\in\mathbb{R}^{m\times r},V\in\mathbb{R}^{n\times r}} \mathbf{E}_{x\sim\mathcal{X}}\left[\mathbf{E}_{b\sim\mathcal{D}}\left[\left\|\left(U_{:b}V_{:b}^{\top}-\tilde{\Sigma}\right)\tilde{V}^{\top}x\right\|^{2}\right]\right].$$

Let $k = \operatorname{rank}(\tilde{\Sigma}) = \operatorname{rank}(A) \leq r$ and $z = \tilde{V}^{\top}x$. Furthermore, let $g = \tilde{\Sigma}z$ for any $z \in \mathbb{R}^{n}$, then $g_{k+1:} = \vec{0}$. This, combined with the nested structure of the optimization problem, implies that the optimal solution for $U$ has to be of the form $u_{i,k+1:} = \vec{0}$ for all interesting (non-zero mapping) directions, i.e., there exists $x \in \mathcal{X}$ such that $v_{i}^{\top}\tilde{V}^{\top}x \neq 0$. These are the only interesting solutions since the case where for all $x \in \mathcal{X} : v_{i}^{\top}\tilde{V}^{\top}x = 0$ yields zero mapping on $\mathcal{X}$, which is not of interest and could be dropped, e.g., using group lasso penalty discussed in the main part. Therefore, to solve the original problem, we could first solve the following problem

$$\min_{U\in\mathbb{R}^{k\times r},V\in\mathbb{R}^{n\times r}} \mathbf{E}_{z\sim\mathcal{X}}\left[\mathbf{E}_{b\sim\mathcal{D}}\left[\left\|\left(U_{:k,:b}V_{:b}^{\top}-\tilde{\Sigma}_{:k,:}\right)z\right\|^{2}\right]\right]$$

and then reconstruct the corresponding solution of the original problem by appending zeros to the resulting matrix $U$. By a similar argument, we can argue that for all non-zero mapping directions, it has to be the case that $v_{i,k+1:} = \vec{0}$. Therefore, solving the original minimization reduces to

$$\min_{U\in\mathbb{R}^{k\times r},V\in\mathbb{R}^{k\times r}} \mathbf{E}_{z\sim\mathcal{X}}\left[\mathbf{E}_{b\sim\mathcal{D}}\left[\left\|\left(U_{:b}V_{:b}^{\top}-\tilde{\Sigma}_{:k,:k}\right)z_{:k}\right\|^{2}\right]\right].$$

This can be further simplified using reparametrization $V^{\top} \to V^{\top}\tilde{\Sigma}_{:k,:k}^{-1}$, which leads to

$$\min_{U\in\mathbb{R}^{k\times r},V\in\mathbb{R}^{k\times r}} \mathbf{E}_{z\sim\mathcal{X}}\left[\mathbf{E}_{b\sim\mathcal{D}}\left[\left\|\left(U_{:b}V_{:b}^{\top}-I_{k}\right)\tilde{\Sigma}_{:k,:k}z_{:k}\right\|^{2}\right]\right], \tag{8}$$

where $I_{k}$ is $k \times k$ identity. If $\mathcal{X}$ is centred around zero, then $\tilde{\Sigma}_{:k,:k}z_{:k}$ is also centred around zero, and the above problem is up to scaling equivalent to PCA of $\tilde{\Sigma}_{:k,:k}z_{:k}$ as shown by Rippel et al. [42]. Since $\tilde{\Sigma}$ is a diagonal matrix with only $k \times k$ non-zero upper left sub-matrix, therefore, PCA on $\tilde{\Sigma}_{:k,:k}z_{:k}$ is equivalent to PCA on $\tilde{\Sigma}z$ by appending zeros to the obtained principal component vectors. Thus, we can write an equivalent formulation

$$\min_{U\in\mathbb{R}^{m\times r},V\in\mathbb{R}^{n\times r}} \mathbf{E}_{z\sim\mathcal{X}}\left[\mathbf{E}_{b\sim\mathcal{D}}\left[\left\|\left(U_{:b}V_{:b}^{\top}-I\right)\tilde{\Sigma}\tilde{V}^{\top}x\right\|^{2}\right]\right].$$

Furthermore, let $\bar{U}, \bar{V}$ belong to the set of optimal solutions of problem (7). Then $U^{\star} = \tilde{U}^{\top}\bar{U}$, $V^{\star} = \tilde{V}^{\top}\bar{V}$ belong to the set of optimal solutions of problem (6). This can be proved by reversing our construction and ignoring scaling since (7) is scaling invariant.

For the case $\mathcal{X}$ is a uniform distribution on the unit ball, we have $\tilde{\Sigma}_{:k,:k}z_{:k}$ is a $k$-dimensional ellipsoid with principal axes being standard basis vectors $\{e_{i}\}_{i=1}^{k}$, where the length of the axes is given by ordered singular values, i.e., the first basis vector corresponds to the largest singular vector. Therefore, its principal component vectors correspond to the basis vectors. Following our construction, one can see that the solution to the original problems leads to truncated SVD up to the scaling factor.

For the case $A$ is an identity, we have $k = r = m = m$, $\tilde{\Sigma}$ is an identity, and $\tilde{U} = \tilde{V}$. Under this setting, the principal component vectors obtained from (8) corresponds to principal component vectors of $\mathcal{X}$ in basis given by columns of $\tilde{U}$. Similarly, as in the previous case, reversing the transformations to return back to the original problem, we conclude that the optimal solution of the original problem corresponds to principal component vectors of $\mathcal{X}$ since we reverse the transformation by $\tilde{U}^{\top}$. $\qquad\square$

## E Experimental setup

### E.1 Datasets

**MNIST.** The MNIST dataset [29] is a database of 28×28 greyscale handwritten digits, with a training set of 60k examples and a test set of 10k samples.

**CIFAR-10.** The CIFAR10 dataset [25] is a computer vision dataset that consists of 32×32 RGB images classified into 10 labels. It is split into 50k training images and 10k test images which are balanced across labels.

**WMT16.** The WMT dataset from statmt is machine translation dataset, spanning news commentaries and parliament proceedings, that aims to investigate the applicability of machine translation techniques when translating between language pairs. Specifically, we focus on the task of German-English language translation of image descriptions, commonly referred to as **Multi30k** [10]. We only utilise the text modality for the translation task. Data is taken straight from `torchtext`.

**TinyImagenet.** The TinyImagenet dataset [28] is a image classification challenge similar to ILSVRC [7]. The task it to classify an 64×64 RGB image among 200 classes, with each class having 500 training samples. The test set contains 10,000 images.

### E.2 Models

**LeNet.** LeNet is a simple convolutional network, introduced by LeCun at al. for recognizing handwritten digits [29]. It consists of a sequence of two convolutional layers, followed by three fully-connected layers. However, we are using a ReLU instead of the initially proposed sigmoid activation. The detailed architecture of the network is depicted in Tab. 5

**ResNet.** ResNet [14] is a deep neural network whose prominent feature is the existence of skip (or residual) connections, that is connections that perform identity mappings merged with the target layer it joins with through summation. Multiple residual blocks are stacked to form the network. The result is an easier to optimise network that offers enhanced accuracy. We use ResNet-18 in our experiments, the architecture of which is depicted in Tab. 6, except for TinyImageNet, where we use ResNet-50.

**Table 5:** Detailed architecture of the LeNet-5 architecture used in our experiments. Each convolution and linear layer is followed by a ReLU activation that is ommitted from the table. The shapes for convolution layers follows $(m, n, k, k)$.

| Parameter | Shape | Layer hyper-parameter |
|---|---|---|
| layer1.conv1.weight | $1 \times 6 \times 5 \times 5$ | stride:1;padding:1 |
| pooling.max | N/A | kernel size:2;stride:1;dilation:1 |
| layer2.conv2.weight | $6 \times 16 \times 5 \times 5$ | stride:1;padding:0;dilation:1 |
| pooling.max | N/A | kernel size:2;stride:2 |
| layer3.fc1.weight | $256 \times 120$ | N/A |
| layer4.fc2.weight | $120 \times 84$ | N/A |
| layer5.fc3.weight | $84 \times 10$ | N/A |

**Table 6:** The hybrid ResNet architecture for the CIFAR-10 and TinyImageNet datasets used in the experiments.

| Layer Name | ResNet-18 | ResNet-50 |
|---|---|---|
| conv1 | 3×3, 64, stride 1, padding 1 | 7×7, 64, stride 2, padding 1 |
| conv2_x | $\begin{bmatrix} 3\times3, 64 \\ 3\times3, 64 \end{bmatrix} \times 2$ | 3×3 maxpool, stride 2 $\begin{bmatrix} 1\times1, 64 \\ 3\times3, 64 \\ 1\times1, 256 \end{bmatrix} \times 3$ |
| conv3_x | $\begin{bmatrix} 3\times3, 128 \\ 3\times3, 128 \end{bmatrix} \times 2$ | $\begin{bmatrix} 1\times1, 128 \\ 3\times3, 128 \\ 1\times1, 512 \end{bmatrix} \times 4$ |
| conv4_x | $\begin{bmatrix} 3\times3, 256 \\ 3\times3, 256 \end{bmatrix} \times 2$ | $\begin{bmatrix} 1\times1, 256 \\ 3\times3, 256 \\ 1\times1, 1024 \end{bmatrix} \times 6$ |
| conv5_x | $\begin{bmatrix} 3\times3, 512 \\ 3\times3, 512 \end{bmatrix} \times 2$ | $\begin{bmatrix} 1\times1, 512 \\ 3\times3, 512 \\ 1\times1, 2048 \end{bmatrix} \times 3$ |
| | Avg Pool, 10-dim FC, SoftMax | Avg Pool, 20-dim FC, SoftMax |

**VGG.** VGG [47] is a also a convolutional network that leverages smaller 3×3 convolutions that enables deeper architecture than before. For our experiments we are using VGG-19, the architecture of which is depicted in Tab. 7.

**Transformers.** The transformer architecture [51] has been lately revolutionising deep learning. Based on the notion of self-attention, for each input token, it produces a weighted combination of other relevant tokens weighed by the attention weight. Each attention unit has three weight matrices, namely $W_Q$, $W_K$, $W_V$, for query, key and value weights respectively producing the equivalent vectors. Attention is defined as the scaled dot product between key and query. For our translation task, we use the architecture depicted in Tab. 9.

**Table 7:** Detailed architecture of the VGG-19 architecture used in our experiments. There is a BatchNorm layer followed by a ReLU activation (omitted in the table) after each convolution layer. The shapes for convolution layers follows $(m, n, k, k)$.

| Parameter | Shape | Layer hyper-parameter |
|---|---|---|
| **layer1.conv1.weight** | $3 \times 64 \times 3 \times 3$ | stride:1;padding:1 |
| **layer2.conv2.weight** | $64 \times 64 \times 3 \times 3$ | stride:1;padding:1 |
| **pooling.max** | N/A | kernel size:2;stride:2 |
| **layer3.conv3.weight** | $64 \times 128 \times 3 \times 3$ | stride:1;padding:1 |
| **layer4.conv4.weight** | $128 \times 128 \times 3 \times 3$ | stride:1;padding:1 |
| **pooling.max** | N/A | kernel size:2;stride:2 |
| **layer5.conv5.weight** | $128 \times 256 \times 3 \times 3$ | stride:1;padding:1 |
| **layer6.conv6.weight** | $256 \times 256 \times 3 \times 3$ | stride:1;padding:1 |
| **layer7.conv7.weight** | $256 \times 256 \times 3 \times 3$ | stride:1;padding:1 |
| **layer8.conv8.weight** | $256 \times 256 \times 3 \times 3$ | stride:1;padding:1 |
| **pooling.max** | N/A | kernel size:2;stride:2 |
| **layer9.conv9.weight** | $256 \times 512 \times 3 \times 3$ | stride:1;padding:1 |
| **layer10.conv10.weight** | $512 \times 512 \times 3 \times 3$ | stride:1;padding:1 |
| **layer11.conv11.weight** | $512 \times 512 \times 3 \times 3$ | stride:1;padding:1 |
| **layer12.conv12.weight** | $512 \times 512 \times 3 \times 3$ | stride:1;padding:1 |
| **pooling.max** | N/A | kernel size:2;stride:2 |
| **layer13.conv13.weight** | $512 \times 512 \times 3 \times 3$ | stride:1;padding:1 |
| **layer14.conv14.weight** | $512 \times 512 \times 3 \times 3$ | stride:1;padding:1 |
| **layer15.conv15.weight** | $512 \times 512 \times 3 \times 3$ | stride:1;padding:1 |
| **layer16.conv16.weight** | $512 \times 512 \times 3 \times 3$ | stride:1;padding:1 |
| **pooling.avg** | N/A | kernel size:2 |
| **classifier.weight** | $512 \times 10$ | N/A |
| **classifier.bias** | 10 | N/A |

**Table 8:** Detailed information of the encoder layer in the Transformer architecture in our experiment

| Parameter | Shape | Hyper-param. |
|---|---|---|
| **embedding** | $9521 \times 512$ | padding index: 1 |
| **positional encoding** | N/A | N/A |
| **dropout** | N/A | $p = 0.1$ |
| **encoder.self-attention.wq($W^Q$)** | $512 \times 512$ | N/A |
| **encoder.self-attention.wk($W^K$)** | $512 \times 512$ | N/A |
| **encoder.self-attention.wv($W^V$)** | $512 \times 512$ | N/A |
| **encoder.self-attention.wo($W^O$)** | $512 \times 512$ | N/A |
| **encoder.self-attention.dropout** | N/A | $p = 0.1$ |
| **encoder.self-attention.layernorm** | 512 | $\varepsilon = 10^{-6}$ |
| **encoder.ffn.layer1** | $512 \times 2048$ | N/A |
| **encoder.ffn.layer2** | $2048 \times 512$ | N/A |
| **encoder.layernorm** | 512 | $\varepsilon = 10^{-6}$ |
| **dropout** | N/A | $p = 0.1$ |

**Table 9:** Detailed information of the decoder layer in the Transformer architecture in our experiment

| Parameter | Shape | Hyper-param. |
|---|---|---|
| **embedding** | $9521 \times 512$ | padding index: 1 |
| **positional encoding** | N/A | N/A |
| **dropout** | N/A | $p = 0.1$ |
| **decoder.self-attention.wq($W^Q$)** | $512 \times 512$ | N/A |
| **decoder.self-attention.wk($W^K$)** | $512 \times 512$ | N/A |
| **decoder.self-attention.wv($W^V$)** | $512 \times 512$ | N/A |
| **decoder.self-attention.wo($W^O$)** | $512 \times 512$ | N/A |
| **decoder.self-attention.dropout** | N/A | $p = 0.1$ |
| **decoder.self-attention.layernorm** | 512 | $\varepsilon = 10^{-6}$ |
| **decoder.enc-attention.wq($W^Q$)** | $512 \times 512$ | N/A |
| **decoder.enc-attention.wk($W^K$)** | $512 \times 512$ | N/A |
| **decoder.enc-attention.wv($W^V$)** | $512 \times 512$ | N/A |
| **decoder.enc-attention.wo($W^O$)** | $512 \times 512$ | N/A |
| **decoder.enc-attention.dropout** | N/A | $p = 0.1$ |
| **decoder.enc-attention.layernorm** | 512 | $\varepsilon = 10^{-6}$ |
| **decoder.ffn.layer1** | $512 \times 2048$ | N/A |
| **decoder.ffn.layer2** | $2048 \times 512$ | N/A |
| **encoder.layernorm** | 512 | $\varepsilon = 10^{-6}$ |
| **dropout** | N/A | $p = 0.1$ |

### E.3 Hyperparameter selection

**LeNet.** We use a standard configuration that is commonly employed for training LeNet models — a step size of 0.01, a momentum of 0.9, and no weight decay. We train for a total of 20 epochs.

**VGG and ResNet-18.** Similarly, we use a standard configuration that is commonly employed for training VGG and ResNet-18 models — a step size of 0.01, a momentum of 0.9, weight decay of $1e^{-4}$, and a learning schedule with step size reductions by a factor of 10 at epochs 150 and 250. We train for a total of 300 epochs.

**ResNet-50.** Similarly, we use a standard configuration that is commonly employed for training ResNet-50 models — a step size of 0.01, a momentum of 0.9, weight decay of $1e^{-4}$, and a learning schedule with step size reductions by a factor of 10 at epochs 30 and 60. We train for a total of 90 epochs.

**Transformers.** For the Transformer model, we use the Adam optimizer with an initial learning rate at $0.001$, $\beta s = (0.9, 0.98), \varepsilon = 10^{-8}$ batch size at $256$. We also conduct gradient norm clipping with norm bound at $0.25$. The entire training takes $400$ epochs. For the vanilla warm-up training, we use warm-up epoch $E_{wu} = 10$. We enable label smoothing, weight sharing for the source and target word embedding, and weight sharing between target word embedding and the last dense layer. The learning rate schedule follows directly from the one proposed [51].

## E.4 Deciding against decomposition

During inference, if the rank of a given layer is so large that keeping it as a non-decomposed layer is more efficient, we opt not to decompose that particular layer.

# F Extended evaluation

## F.1 MAESTRO **recovers correct ordering**

In the main text, we pointed out that SVD fails to consider data. Indeed, even in the case of linear NN, the acquired singular vectors may exhibit incorrect ordering. To illustrate this problem, we provide a simple example in which we use a matrix $A$ with a rank of $3$. We organize the dataset $\mathcal{X}$ such that the third singular vector has the highest importance, followed by the second and then the first singular vector in decreasing order of significance. It is clear that SVD doesn't consider the data, and as a result, it cannot comprehend this behavior. Below (in Fig. 5), we demonstrate how MAESTRO is able to correctly discern the order.

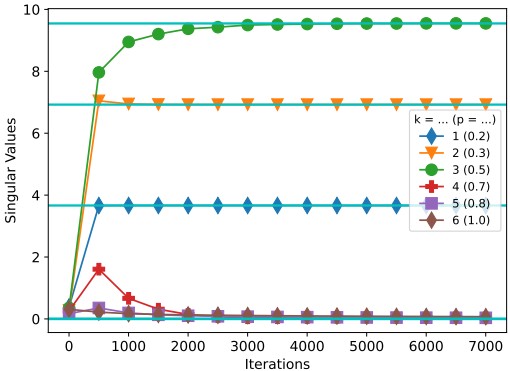

**Figure 5:** Verification that MAESTRO recovers correct order of importance. Target mapping is of rank 3, and the dataset is constructed in such a way that the singular vectors have reversed the order of importance. $p$ and $k$ stand for relative and actual rank, respectively.

## F.2 Training behaviour of MAESTRO

For completeness, we also include an extended version of Fig. 3 from the main paper, where we presented the training dynamics for MAESTRO. Fig.6, 7 and 8 present similar plots, but across both MNIST and CIFAR-10. Specifically, Fig. 6 illustrates the evolution of total rank throughout the training steps. We observe that the ranks are pruned incrementally. This aligns with the observations made during Pufferfish [53] training, where the authors suggested warm-start training with full

precision to enhance the final model performance. In our case, the necessity to implement this heuristic is avoided, as MAESTRO prunes rank automatically. Fig. 7 demonstrates the ranks across layers post-training. An intriguing trend is observed: the ranks are nested for increasing $\lambda_{gl}$, suggesting a potential inherent ordering of ranks not only within each layer but also possibly a global one. We provide a preliminary exploration of this fascinating pattern in the subsequent section and intend to probe it more deeply in future studies. We believe this may enhance the rank selection and sampling process. Finally, Fig. 8 portrays the evolution of the training loss. Our premise that sampling does not negatively affect training is validated by empirical performance.

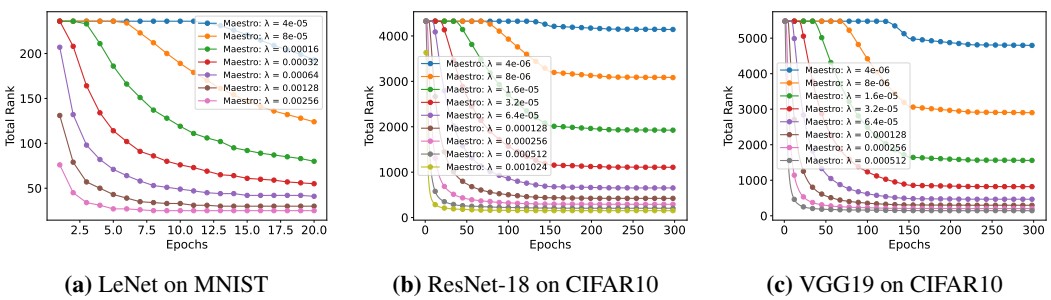

(a) LeNet on MNIST      (b) ResNet-18 on CIFAR10      (c) VGG19 on CIFAR10

**Figure 6:** Total rank ($\sum_{i=1}^{d} r_i$) progression during training.

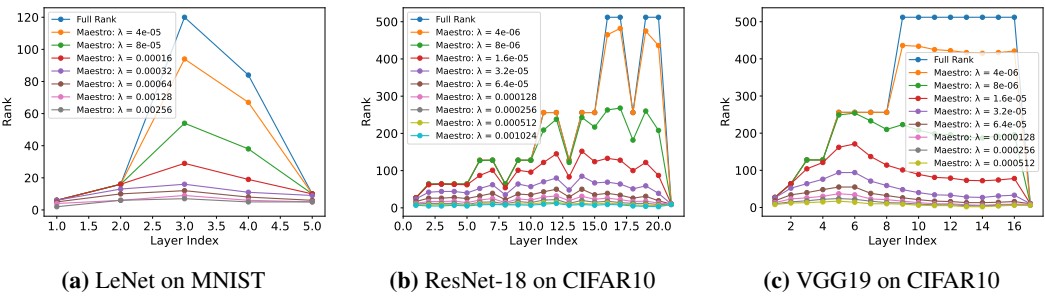

(a) LeNet on MNIST      (b) ResNet-18 on CIFAR10      (c) VGG19 on CIFAR10

**Figure 7:** Ranks $r_i$'s across different layers after training.

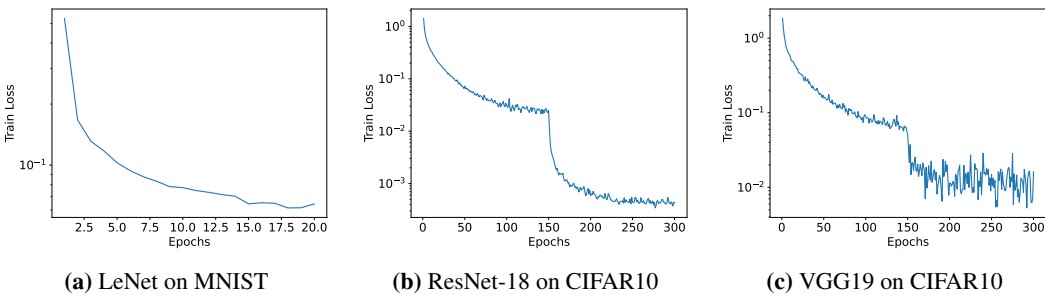

(a) LeNet on MNIST      (b) ResNet-18 on CIFAR10      (c) VGG19 on CIFAR10

**Figure 8:** Convergence of MAESTRO with $\lambda_{gl} = 0$.

### F.3    Model size-accuracy trade-off at training and deployment time

In addition to the original illustrations, we present an extended interpretation of Fig. 4, where we depict diverse strategies to maintain a balance between model size and accuracy in the process of model training and deployment. In Fig. 9, we demonstrate the effective pruning of MAESTRO ($\lambda_{gl} = 0$) for deployment, utilizing the greedy search methodology discussed in Section 3.4. This is juxtaposed with the greedy pruning of a model not originally factorized but later factorized through SVD. Our findings reveal that this straightforward baseline does not match the performance of MAESTRO's learned decomposition, leading to a considerable performance drop.

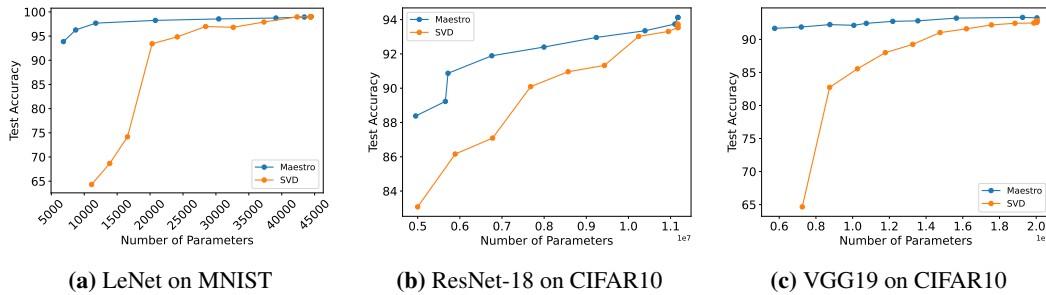

**(a)** LeNet on MNIST  **(b)** ResNet-18 on CIFAR10  **(c)** VGG19 on CIFAR10

**Figure 9:** Accuracy-latency trade-off comparing MAESTRO ($\lambda_{\mathrm{gl}=0}$) and SVD.

Subsequently, Fig. 10 displays the end accuracy and the count of model parameters corresponding to various hierarchical group lasso penalties. This results in an optimal compromise between latency and accuracy for both the training and inference stages. It's worth noting, though, that each model was trained separately, in contrast to greedy pruning, which demands just a single training round. Additionally, we scrutinize the training expense for each model illustrated in Fig. 10, the results of which are exhibited in Tables 10, 11, 12, 13 and 14, where we display and the final accuracy of the model, MACs and the number of parameters for inference, and relative total training cost in terms of the number of model parameters and MACs compared to the non-factorized model. Interestingly, smaller models are not only advantageous in terms of inference efficiency, but they can also be trained at a small portion of the cost required by full-rank models. On the downside, the smallest models cause a non-negligible reduction in performance.

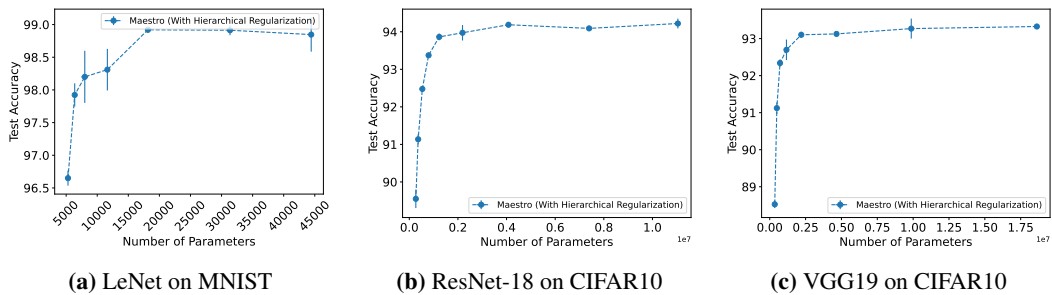

**(a)** LeNet on MNIST  **(b)** ResNet-18 on CIFAR10  **(c)** VGG19 on CIFAR10

**Figure 10:** Impact of hierarchical group lasso on the accuracy-memory trade-off. Exact values are provided in Tables 10, 11 and 12, respectively.

**Table 10:** LeNet performance on MNIST for different regularization parameters. The last column in the table displays the relative total training cost in terms of the number of Multiply-Accumulate operations (MACs) and model parameters, compared to the non-factorized model.

| Variant | Acc. (%) | MACs (Inf.) | Params. (Inf.) | Rel. MACs / Params. (Train.) |
|---|---|---|---|---|
| Non-Factorized | $98.99_{\pm 0.09}$ | $281640_{\pm 0}$ ($1.00\times$) | $44426_{\pm 0}$ ($1.00\times$) | $1.00\times$ / $1.00\times$ |
| MAESTRO ($\lambda_{gp}=0.$) | $99.06_{\pm 0.09}$ | $281640_{\pm 0}$ ($1.00\times$) | $44426_{\pm 0}$ ($1.00\times$) | $1.14\times$/ $1.49\times$ |
| MAESTRO ($\lambda_{gp}=8e^{-5}$) | $98.91_{\pm 0.09}$ | $268577_{\pm 389}$ ($0.95\times$) | $31363_{\pm 0}$ ($0.71\times$) | $1.08\times$/ $1.14\times$ |
| MAESTRO ($\lambda_{gp}=16e^{-5}$) | $98.92_{\pm 0.05}$ | $255369_{\pm 217}$ ($0.91\times$) | $44426_{\pm 217}$ ($0.41\times$) | $1.06\times$/ $0.80\times$ |
| MAESTRO ($\lambda_{gp}=32e^{-5}$) | $98.31_{\pm 0.39}$ | $237084_{\pm 6268}$ ($0.84\times$) | $18155_{\pm 271}$ ($0.26\times$) | $0.93\times$/ $0.53\times$ |
| MAESTRO ($\lambda_{gp}=64e^{-5}$) | $98.20_{\pm 0.49}$ | $178165_{\pm 19098}$ ($0.63\times$) | $7996_{\pm 662}$ ($0.18\times$) | $0.77\times$/ $0.33\times$ |
| MAESTRO ($\lambda_{gp}=128e^{-5}$) | $97.92_{\pm 0.22}$ | $131789_{\pm 8965}$ ($0.47\times$) | $6375_{\pm 77}$ ($0.14\times$) | $0.54\times$/ $0.21\times$ |
| MAESTRO ($\lambda_{gp}=256e^{-5}$) | $96.65_{\pm 0.14}$ | $99969_{\pm 6252}$ ($0.35\times$) | $5293_{\pm 214}$ ($0.12\times$) | $0.39\times$/ $0.14\times$ |

Lastly, we delve deeper into the observation of nested ranks with increasing $\lambda_{gl}$. Fig. 11 outlines the performance of MAESTRO ($\lambda_{gl}=0$) across various ranks chosen by smaller models MAESTRO ($\lambda_{gl}>0$). We observe that MAESTRO ($\lambda_{gl}=0$) delivers impressive results—for example, we can reduce its parameters by 10x for VGG while preserving an accuracy of $87.7\%$ without any fine-tuning simply by leveraging rank structure from separate runs. For LeNet, a reduction in model size by a factor of three is achievable without sacrificing accuracy. Last, for ResNet-18 the reduction is $1.7\times$. As highlighted earlier, we aim to delve deeper into this subject in future studies.

**Table 11:** ResNet-18 performance on CIFAR10 for different regularization parameters. The last column in the table displays the relative total training cost in terms of the number of Multiply-Accumulate operations (MACs) and model parameters, compared to the non-factorized model.

| Variant | Acc. (%) | GMACs (Inf.) | Params. (M) (Inf.) | Rel. MACs / Params. (Train.) |
|---|---|---|---|---|
| Non-Factorized | $93.86_{\pm 0.20}$ | $0.56_{\pm 0}$ $(1.00\times)$ | $11.2_{\pm 0}$ $(1.00\times)$ | $1.00\times$ / $1.00\times$ |
| MAESTRO ($\lambda_{gp} = 0.$) | $94.04_{\pm 0.10}$ | $0.56_{\pm 0}$ $(1.00\times)$ | $11.2_{\pm 0}$ $(1.00\times)$ | $1.10\times$ / $1.13\times$ |
| MAESTRO ($\lambda_{gp} = 4e^{-6}$) | $94.22_{\pm 0.16}$ | $0.55_{\pm 0.0047}$ $(1.00\times)$ | $11.1_{\pm 0.030}$ $(0.99\times)$ | $1.09\times$ / $1.10\times$ |
| MAESTRO ($\lambda_{gp} = 8e^{-6}$) | $94.09_{\pm 0.01}$ | $0.49_{\pm 0.0002}$ $(0.89\times)$ | $7.41_{\pm 0.004}$ $(0.66\times)$ | $1.00\times$ / $0.85\times$ |
| MAESTRO ($\lambda_{gp} = 16e^{-6}$) | $94.19_{\pm 0.07}$ | $0.39_{\pm 0.0008}$ $(0.70\times)$ | $4.08_{\pm 0.020}$ $(0.37\times)$ | $0.83\times$ / $0.58\times$ |
| MAESTRO ($\lambda_{gp} = 32e^{-6}$) | $93.97_{\pm 0.25}$ | $0.25_{\pm 0.0013}$ $(0.45\times)$ | $2.19_{\pm 0.007}$ $(0.20\times)$ | $0.60\times$ / $0.36\times$ |
| MAESTRO ($\lambda_{gp} = 64e^{-6}$) | $93.86_{\pm 0.11}$ | $0.15_{\pm 0.0006}$ $(0.27\times)$ | $1.23_{\pm 0.004}$ $(0.11\times)$ | $0.39\times$ / $0.22\times$ |
| MAESTRO ($\lambda_{gp} = 128e^{-6}$) | $93.37_{\pm 0.07}$ | $0.094_{\pm 0.0006}$ $(0.17\times)$ | $0.79_{\pm 0.009}$ $(0.07\times)$ | $0.25\times$ / $0.13\times$ |
| MAESTRO ($\lambda_{gp} = 256e^{-6}$) | $92.48_{\pm 0.04}$ | $0.064_{\pm 0.0002}$ $(0.12\times)$ | $0.54_{\pm 0.006}$ $(0.05\times)$ | $0.16\times$ / $0.08\times$ |
| MAESTRO ($\lambda_{gp} = 512e^{-6}$) | $91.14_{\pm 0.16}$ | $0.044_{\pm 0.0004}$ $(0.08\times)$ | $0.37_{\pm 0.007}$ $(0.03\times)$ | $0.11\times$ / $0.05\times$ |
| MAESTRO ($\lambda_{gp} = 1024e^{-6}$) | $89.55_{\pm 0.30}$ | $0.032_{\pm 0.0002}$ $(0.06\times)$ | $0.27_{\pm 0.007}$ $(0.02\times)$ | $0.07\times$ / $0.03\times$ |

**Table 12:** VGG19 performance on CIFAR10 for different regularization parameters. The last column in the table displays the relative total training cost in terms of the number of Multiply-Accumulate operations (MACs) and model parameters, compared to the non-factorized model.

| Variant | Acc. (%) | GMACs (Inf.) | Params. (M) (Inf.) | Rel. MACs / Params. (Train.) |
|---|---|---|---|---|
| Non-Factorized | $92.94_{\pm 0.17}$ | $0.40_{\pm 0}$ $(1.00\times)$ | $20_{\pm 0}$ $(1.00\times)$ | $1.00\times$ / $1.00\times$ |
| MAESTRO ($\lambda_{gp} = 0.$) | $93.06_{\pm 0.17}$ | $0.40_{\pm 0}$ $(1.00\times)$ | $20_{\pm 0}$ $(1.00\times)$ | $1.10\times$ / $1.12\times$ |
| MAESTRO ($\lambda_{gp} = 4e^{-6}$) | $93.33_{\pm 0.08}$ | $0.39_{\pm 0.0017}$ $(0.97\times)$ | $18.8_{\pm 0}$ $(0.94\times)$ | $1.06\times$ / $1.04\times$ |
| MAESTRO ($\lambda_{gp} = 8e^{-6}$) | $93.27_{\pm 0.33}$ | $0.30_{\pm 0.0017}$ $(0.76\times)$ | $9.91_{\pm 0.008}$ $(0.49\times)$ | $0.90\times$ / $0.73\times$ |
| MAESTRO ($\lambda_{gp} = 16e^{-6}$) | $93.13_{\pm 0.07}$ | $0.21_{\pm 0.0014}$ $(0.53\times)$ | $4.66_{\pm 0.052}$ $(0.23\times)$ | $0.69\times$ / $0.46\times$ |
| MAESTRO ($\lambda_{gp} = 32e^{-6}$) | $93.10_{\pm 0.10}$ | $0.13_{\pm 0.0009}$ $(0.33\times)$ | $2.20_{\pm 0.025}$ $(0.11\times)$ | $0.47\times$ / $0.27\times$ |
| MAESTRO ($\lambda_{gp} = 64e^{-6}$) | $92.70_{\pm 0.34}$ | $0.08_{\pm 0.0005}$ $(0.20\times)$ | $1.17_{\pm 0.010}$ $(0.06\times)$ | $0.30\times$ / $0.16\times$ |
| MAESTRO ($\lambda_{gp} = 128e^{-6}$) | $92.34_{\pm 0.12}$ | $0.05_{\pm 0.0005}$ $(0.13\times)$ | $0.72_{\pm 0.002}$ $(0.04\times)$ | $0.19\times$ / $0.09\times$ |
| MAESTRO ($\lambda_{gp} = 256e^{-6}$) | $91.12_{\pm 0.19}$ | $0.04_{\pm 0.0007}$ $(0.09\times)$ | $0.50_{\pm 0.023}$ $(0.02\times)$ | $0.12\times$ / $0.05\times$ |
| MAESTRO ($\lambda_{gp} = 512e^{-6}$) | $88.53_{\pm 0.13}$ | $0.03_{\pm 0.0003}$ $(0.06\times)$ | $0.35_{\pm 0.003}$ $(0.02\times)$ | $0.08\times$ / $0.03\times$ |

**Table 13:** Transformer performance on Multi30k for different regularization parameters. The last column in the table displays the relative total training cost in terms of the number of Multiply-Accumulate operations (MACs) and model parameters, compared to the non-factorized model.

| Variant | Acc. (%) | Ppl. | GMACs (Inf.) | Params. (M) (Inf.) | Rel. MACs / Params. (Train.) |
|---|---|---|---|---|---|
| Non-Factorized | $65.33_{\pm 1.13}$ | $9.85_{\pm 0.10}$ | $1.370_{\pm 0.0000}$ $(1.00\times)$ | $53.9_{\pm 0.000}$ $(1.00\times)$ | $1.00\times$ / $1.00\times$ |
| MAESTRO ($\lambda_{gp} = 0.32$) | $61.30_{\pm 0.26}$ | $12.99_{\pm 0.31}$ | $1.125_{\pm 0.0030}$ $(0.82\times)$ | $45.1_{\pm 0.101}$ $(0.84\times)$ | $1.03\times$ / $1.14\times$ |
| MAESTRO ($\lambda_{gp} = 0.64$) | $63.78_{\pm 0.14}$ | $9.37_{\pm 0.32}$ | $0.957_{\pm 0.0112}$ $(0.70\times)$ | $39.1_{\pm 0.413}$ $(0.73\times)$ | $0.95\times$ / $1.05\times$ |
| MAESTRO ($\lambda_{gp} = 1.28$) | $66.14_{\pm 0.08}$ | $7.02_{\pm 0.17}$ | $0.570_{\pm 0.0088}$ $(0.42\times)$ | $25.3_{\pm 0.315}$ $(0.47\times)$ | $0.75\times$ / $0.86\times$ |
| MAESTRO ($\lambda_{gp} = 2.56$) | $66.08_{\pm 0.09}$ | $6.90_{\pm 0.07}$ | $0.248_{\pm 0.0032}$ $(0.18\times)$ | $13.8_{\pm 0.113}$ $(0.26\times)$ | $0.47\times$ / $0.58\times$ |
| MAESTRO ($\lambda_{gp} = 5.12$) | $57.70_{\pm 0.13}$ | $13.97_{\pm 0.43}$ | $0.123_{\pm 0.0002}$ $(0.9\times)$ | $9.3_{\pm 0.001}$ $(0.17\times)$ | $0.28\times$ / $0.39\times$ |

**Table 14:** ResNet50 performance on Tiny-Imagenet-200 for different regularization parameters. The last column in the table displays the relative total training cost in terms of the number of Multiply-Accumulate operations (MACs) and model parameters, compared to the non-factorized model.

| Variant | Acc. (%) | GMACs (Inf.) | Params. (M) (Inf.) | Rel. MACs / Params. (Train.) |
|---|---|---|---|---|
| Non-Factorized | $61.74_{\pm 0.27}$ | $5.19_{\pm 0.0000}$ $(1.00\times)$ | $23.9_{\pm 0.000}$ $(1.00\times)$ | $1.22\times$ / $1.22\times$ |
| MAESTRO ($\lambda_{gp} = 0.$) | $61.05_{\pm 0.09}$ | $5.19_{\pm 0.0000}$ $(1.00\times)$ | $23.9_{\pm 0.000}$ $(1.00\times)$ | $1.21\times$ / $1.20\times$ |
| MAESTRO ($\lambda_{gp} = 4e^{-5}$) | $60.13_{\pm 0.34}$ | $4.72_{\pm 0.0013}$ $(0.91\times)$ | $18.8_{\pm 0.017}$ $(0.79\times)$ | $0.81\times$ / $0.69\times$ |
| MAESTRO ($\lambda_{gp} = 8e^{-5}$) | $59.20_{\pm 0.40}$ | $3.01_{\pm 0.0064}$ $(0.58\times)$ | $9.64_{\pm 0.023}$ $(0.40\times)$ | $0.00\times$ / $0.00\times$ |
| MAESTRO ($\lambda_{gp} = 16e^{-5}$) | $58.35_{\pm 0.40}$ | $1.49_{\pm 0.0142}$ $(0.29\times)$ | $4.48_{\pm 0.022}$ $(0.19\times)$ | $0.61\times$ / $0.54\times$ |
| MAESTRO ($\lambda_{gp} = 32e^{-5}$) | $56.52_{\pm 0.08}$ | $0.72_{\pm 0.0022}$ $(0.14\times)$ | $2.25_{\pm 0.013}$ $(0.09\times)$ | $0.51\times$ / $0.47\times$ |

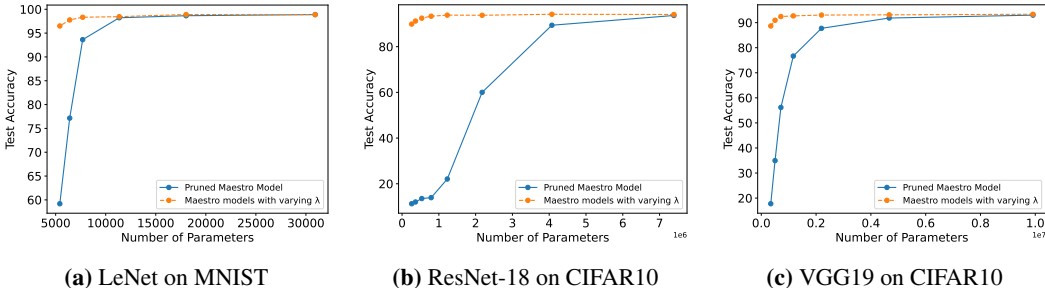

**(a)** LeNet on MNIST        **(b)** ResNet-18 on CIFAR10        **(c)** VGG19 on CIFAR10

**Figure 11:** MAESTRO with progressive pruning to showcase nested rank importance structure. The original model corresponds to an evaluation in Fig. 10, and pruned models are based on MAESTRO with $\lambda_{gl} = 0$, and they are pruned using the same ranks as selected by MAESTRO with $\lambda_{gl} > 0$.

