# OpenReview forum: "Maestro: Uncovering Low-Rank Structures via Trainable Decomposition"
_NeurIPS.cc/2023/Workshop/WANT — WANT@NeurIPS 2023 Poster_

### Official Review · Reviewer_x31F · 2023-10-24
**The paper introduces a technique called "Maestro" to train low rank layers progressively by incorporating ordered dropout into layers' weights and eliminating redundant ranks. The approach is supported by theoretical analysis and backed up by empirical evidence through experimental results.**

**Confidence:** 3

**Review:**

[Strengths]

The paper introduces a novel approach for low rank layer learning. The main claims are supported by experimental results, which add credibility to the proposed method.

[Weaknesses]

One potential weakness of the paper is that it only considers one approach for comparison with the transformer model. Including multiple approaches for comparison would provide a stronger evaluation of the proposed technique.

[Novelty and significance]

The proposed approach is deemed interesting and novel. The idea of setting individual optimal ranks for each layer is reasonable and allows for effective rank extraction when combined with Hierarchical Group Lasso. The use of rank sampling during training enhances computational efficiency and has shown promising empirical results. The theoretical support provided for the proposed technique adds to its significance.

[Technical quality and empirical evaluation]

The paper is well-written and presents experimental results that generally support the main claims. However, there is a lack of comparison for the transformer model. The inclusion of a quantization approach could further enhance empirical evaluation.

---

### Official Review · Reviewer_Zr5e · 2023-10-25
**Improved low rank approximation by making it trainable and data aware**

**Confidence:** 3

**Review:**

This work proposes an interesting idea of incorporating low rank approximation in training. One significant benefit is by doing so, matrices that are to be factorized will be more friendly to low rank approximation. Thus for in-domain data, this approach should work better than standard post low rank approximation approaches such as SVD.

The paper is well written, the explanation of the approach is clear, and the comparison with other approaches are comprehensive. One potential issue that I can think of is at the test time the proposed approach might be worse if the data is completely out of domain, which might happen a lot in reality. However, certain regularization might help to mitigate this issue.

---

### Official Review · Reviewer_ouEG · 2023-10-25
**The paper designs efficient low-rank models and propose MAESTRO, a framework for trainable low-rank layers**

**Confidence:** 5

**Review:**

Instead of regularly applying a priori decompositions such as SVD, a low-rank structure is embedded in the training process through a generalized variant of ordered dropout. An analysis is also provided which shows that the method recovers the SVD decomposition of linear mapping under certain conditions. The work also addresses the selection of the rank per layer and the trade-off between the efficiency and the rank for a given layer.

Pros
- Topic interesting: MAESTRO formulation is directly built into training and tailored to obtain the best low-rank decomposition seemingly better than standard SVD both for efficiency and effectiveness.
- The paper is globally decently written though some parts are not sufficiently clear.

Cons
-  Contribution is incremental: the use of SVD decomposition (and more generally tensor decomposition) either in a non-parametric or parametric way (i.e., as a part of network design) is a well studied topic (in deep neural networks, etc).
- There are many issues in the notation and the math used in different equations (see below).

Other remarks
- Check again eq 1, tilde u, tilde v used a few lines earlier, etc ... besides, where are the singular values ...
- Not clear why the sum in eq2 is outside the F-norm: is eq2 an upper bound of eq1? this needs a clarification and better smoothing.
- In the notation of eq3, x is the only one taken from cal X (and not y).
- In the sentence "We propose to decompose each layer independently to uncover its – potentially different – optimal rank
for deployment." -> to what extent it is beneficial to achieve this decomposition independently for each layer ? what is the benefit besides heterogeneous ranks?
- The sentence "Wo are the other weights that we do not decompose" should be further clarified.
- In line 3 of algorithm 1, why not sampling jointly through layers?  this should be carefully discussed.
- The gain is marginal on CIFAR10.

---

### Meta-Review · Area_Chair_XJ2W · 2023-10-27

**Recommendation:** Accept (Poster)
**Confidence:** 4

**Metareview:**

**Strengths:**
* The reviewers found the paper to be well-written with clear explanations. Some parts (notation, math) could, however, use improvement.
* Paper addresses a relevant problem.
* Most of the reviewers found the contributions interesting and novel.

**Weaknesses:**
* Marginal gains on CIFAR-10
* Limited evaluation of Transformer-based models.

The overall sentiment for the submission based on the reviews appears to be positive, although a bit borderline. I recommend acceptance (poster).

---

### Decision · Program_Chairs · 2023-10-28

**Decision:**

Accept (Poster)

**Comment:**

We thank the authors for their time and contribution to WANT and we are pleased to share that after the reviewing process the paper has been accepted. Congratulations! We encourage the authors to consider reviewers' feedback for the improvement of the camera-ready version. We hope to see you in person at the workshop and brainstorm on efficient training research together!